# Regeneration-Associated Transitional State Cells in Pulmonary Fibrosis

**DOI:** 10.3390/ijms23126757

**Published:** 2022-06-17

**Authors:** Mengxia Shen, Ziqiang Luo, Yan Zhou

**Affiliations:** 1Department of Physiology, Xiangya School of Medicine, Central South University, Changsha 410003, China; shenmx0421@163.com (M.S.); luoziqiang@csu.edu.cn (Z.L.); 2Hunan Key Laboratory of Organ Fibrosis, Changsha 410003, China

**Keywords:** pulmonary fibrosis, transitional state cell, alveolar regeneration

## Abstract

Pulmonary fibrosis is a chronic, progressive fibrosing interstitial disease. It is characterized by fibroblast proliferation, myofibroblast activation, and massive extracellular matrix deposition. These processes result in loss of lung parenchyma function. The transdifferentiation of alveolar epithelial type II (AEC2) to alveolar epithelial type I cells (AEC1) plays an important role in the epithelial repair after lung injury. Pulmonary fibrosis begins when this transdifferentiation process is blocked. Several recent studies have found that novel transitional state cells (intermediate states in the transdifferentiation of AEC2 to AEC1) can potentially regenerate the alveolar epithelium surface and promote a repair process. During the AEC2 to AEC1 trans-differentiation process after injury, AEC2 lose their specific markers and become transitional state cells. Furthermore, transdifferentiation of transitional state cells into AEC1 is the critical step for lung repair. However, transitional cells stagnate in the intermediate states in which failure of transdifferentiation to AEC1 may induce an inadequate repair process and pulmonary fibrosis. In this review, we focus on the traits, origins, functions, and activation of signaling pathways of the transitional state cell and its communication with other cells. We also provide a new opinion on pulmonary fibrosis pathogenesis mechanisms and novel therapeutic targets.

## 1. Introduction

Pulmonary fibrosis is a rare and age-related interstitial pneumonia with a high mortality rate. Its cause is unknown. Average life expectancy typically will not exceed 5 years after diagnosis without treatment. In the future, we estimate that the occurrence of pulmonary fibrosis in patients will steadily increase. Currently, Pirfenidone and Nintedanib, both FDA-approved, are the most commonly prescribed anti-fibrotic medications. Unfortunately, these drugs only temporarily relieve symptoms. They fail to reduce the progression of fibrosis [1]. Several studies provide evidence that alveolar epithelial cells play a central role in pulmonary function repair. Animal model studies have highlighted that impaired epithelial regeneration after pulmonary fibrosis is an underlying mechanism of pulmonary fibrotic disease [2]. In healthy lungs, alveolar epithelial cells include type I (AEC1) and type II cells (AEC2). Generally, cuboidal AEC2 cells, a heterogeneous population that maintains widespread proliferation and differentiation for self-renewal, produce surfactants and decrease surface tension. Squamous AEC1 cells cover most of the alveolus surface and mediate gas exchange. AEC2 acts as progenitor cells, transdifferentiating into AEC1 [3,4,5]. When an injury occurs, AEC1 has a great sensitivity to damage, and excessive death causes impaired barrier function. In response to lung damage, AEC1 cells are able to provide an alveolar niche with enhanced flexibility by increasing the number of AEC2 cells. In turn, these generate new AEC1 cells [6].

However, during the process of AEC2 transdifferentiating to AEC1, the failure of AEC1 maturation may lead to an accumulation of cells “stuck” in a transitional state. The transitional state cell lies between the differentiation from AEC2 to AEC1. It is critical to lung regeneration, yet its phenotype is less well understood. Several previous studies have elucidated the cells’ characteristics, including growth arrest, differentiation interruption, and senescence, and the important role they play in alveolar regeneration and organ repair [7]. This transitional state cell may well be the essential factor in fibrosis pathogenesis, as they surround the fibrotic foci, showing similar transcriptional characteristics to fibrogenesis and senescence [8].

The transitional state cell has attracted much scientific attention. Yet, there remains a lack of review papers regarding their role in respiratory diseases, especially pulmonary fibrosis. Therefore, in this review, we focus on the traits, origins, functions, and activation of signaling pathways of the transitional state cell from recent single-cell RNA sequencing (scRNA-seq) analysis data and animal studies. Moreover, we examine the role played by the transitional state cell in lung disease and its effects on mediating lung regeneration to increase the comprehensive understanding of these cells.

## 2. Traits of the Transitional State Cells

In normal lung tissue, there are varieties of identified cell populations, including AEC1, AEC2, club cells, basal cells, ciliated cells, and so on. However, in idiopathic pulmonary fibrosis (IPF), there are multiple novel cell states. These states mediate the regeneration of the alveolar epithelium [9]. Among these states, Adam et al. [10] found a distinctive cell type distinguishable from other epithelial cells in IPF. They designated this intermediate state of transdifferentiation of AEC2 to AEC1 as an aberrant basaloid cell. The aberrant basaloid cell is located in fibrotic lesions in IPF. It possesses characteristics of growth arrest, differentiation interruption and senescence, indicating these aberrant basaloid cells might intensify pulmonary fibrosis [11].

The novel state of the transitional state cell is expressed by multiple markers, including basal cell markers, senescence-related genes, IPF-related molecules, epithelial-mesenchymal transition (EMT)-related markers, and senescence-related genes [12]. By isolating and culturing lung tissue cells in vitro from patients with interstitial lung diseases (ILD), Khan et al. [13] confirmed their homogeneity and similar transcriptome characteristics compared to normal epithelial cells. They found that the transitional state cell expressed basal markers (*Cytokeratin (Krt) 17, TP63, Lamc2*). Therefore, these cells are referred to as “basal-like cells.” Additionally, they express high levels of mesenchymal cell markers (*VIM, FN1, Col1A1*) [14], alveolar markers (*Abca3, Emp2, Hopx*), secretory epithelial cell markers (*Scgb1A1, Muc4*) [15], senescence cell markers (*Ccnd1, Ccnd2, Mdm2*) [16], and IPF molecules (*Ephb2, matrix metallopeptidase 7 [Mmp7]*) [17]. Ting et al. [18] reported that during physiological regeneration, the transitional state cell expresses markers of cell cycle stages in vivo. However, the cells expressed a specific marker of senescence only in IPF. Furthermore, they hypothesized an intriguing situation-specific capability in which AEC2 would be able to temporarily arrest in the transitional state and then spontaneously differentiate into AEC1 after lung injury. However, these cells may permanently reside in the transitional state with senescence markers, losing the potential to differentiate into AEC1 and ultimately hinder or obstruct fibrosis.

The transitional progenitor cell is an abnormal transitional intermediate cell state. It is recognized and accepted that the presence of transitional state cells in lung injury blocks the normal repair progression of alveolar epithelial cells, as discussed in animal studies. Huang et al. [19] integrated the transitional alveolar epithelial state of IPF between humans and mice. They found that human *KRT5^−^/KRT17^+^* cells originate from the dysregulation of AEC2 to AEC1. Furthermore, *KRT5^−^/KRT17^+^* cells may play an important role in alveoli regeneration in SARS-CoV-2 (COVID-19) influenza. Through generating single-cell maps of multiple organs, including those within the pulmonary system, connections between failed tissue regeneration and defective AEC2 differentiation and putative transitional cells have been identified [20,21]. In the future, it will be important to determine the relationship between transitional state cells and alveolar regeneration, given that it is possible that reducing the accumulation of these transitional cells and promoting transformation could contribute to a delayed progression of fibrotic diseases.

## 3. Features of Intermediate Transitional Cells at Different Stages

After lung injury, the plasticity of alveolar epithelium cells is associated with non-gradual transdifferentiation of intermediate transition cells. This process results in a unique, aberrant progenitor cell state. By re-analyzing the scRNA-seq data from murine or human studies, the intermediate progenitor cell population can be divided into several different stages, as shown in Figure 1 [7,11,22,23]. Five stages are summarized, including the *Ctgf* pre-alveolar type-1 transitional cell (*Ctgf* PATs), *Krt8^+^* alveolar differentiation intermediate (*Krt8^+^* ADI), damage-associated transient progenitors (DATPs), *Lgals3* PATs, and pre-AEC1 states [11,19,24]. *Ctgf* PATs are enriched for transforming growth factor beta (*TGF-β*), *Ctgf*, *Clu*, and *Sox4*. *Krt8^+^* ADI is characterized by a marked decrease in *Etv5*, important for AEC2 maintenance, and a corresponding increase in the expression of *p53* pathway components and the *Krt8* peak. DATP cells have a high transcriptomic similarity with *Krt8^+^* ADI. The expression of the *p53* pathway increases in this stage. *Lgals3^+^* PATs are enriched for *Lgals3*, *Csrp1*, and *Cldn18*, and for markers of AEC1 such as *Ager* and *Hopx*, implying that *Lgals3* PAT cells may mature in a stage closer to AEC1. In addition, pre-AEC1 is marked by a decreased level of *Krt8* and activation of specific genes such as *Myh9*, *Itgb6*, and *Ctnnb1*. These contribute to cytoskeleton rearrangement, increase cell contractility and, finally, accelerate the maturation of pre-AEC1.

In general, each subset may exist in differentiation trajectories from AEC2, airway, or club cells to AEC1. There may be an overlap between cellular states. Therefore, it is difficult to distinguish different cellular states. Clarifying the individual subsets of transitional state cells could contribute to fibrotic disease treatments. During lung regeneration, AEC2 differentiation stagnation leads to persistence and accumulation of transitional state alveolar epithelial cells. Therefore, stagnation eventually may aggravate progressive lung fibrosis.

## 4. Origins of Transitional State Cell

Recent research indicates that lung injury in mice and humans undoubtedly promotes the accumulation of transitional state cells. However, no conclusive evidence exists regarding these cells’ origins. Some organoids, injury models, and lineage tracing studies have suggested that these cells might arise from AEC2 [25,26]. Utilizing *Sftpc-creER*, the *R26R-tdTomato* mouse model revealed how AEC2 transitions to AEC1, highlighting that there is specific cell morphology during the differentiation process [11]. Additionally, by using *Sftpc-CreERT2* (AEC2) and *Sox2-CreERT2* (airway) reporter mice, it was found that half of the intermediate transition cells were derived from either AEC2 or airway cells [24]. This finding indicated a dual origin of the transitional state cell. Jiang et al. [27] corroborated the *Krt8^+^* transition state cells’ persistence during ineffectual AEC2-to-AEC1 differentiation. The cell arrest of AEC2 early in differentiation might lead to the failed reconstruction of alveolar architecture after lung injury. It has also been found that not all transitional state cells transformed from AEC2. Airway-resident intrapulmonary *p63^+^* progenitors migrate into the damaged area and proliferate. These progenitor cells show similar cell markers and embryonic origins to tracheal basal cells. However, this stem cell population differs from tracheal basal cells. They serve as fundamental basal cells from the trachea, meaning that airway cells might be a pivotal origin of intrapulmonary *p63^+^* progenitors [28,29]. Lineage-tracing studies with *Sox2-Cre* (airway cell) reporter mice, such as that of Strunz et al. [30], have found that positive cells appeared in the lung after bleomycin-induced pulmonary fibrosis. Therefore, it is believed that the *Krt8^+^* alveolar epithelial progenitors partially originate from the airway.

In addition, there is no clear evidence to confirm whether the transitional state cell expresses the club cell markers. However, club cells serve as a type of regional progenitor cell that repairs the bronchiolar epithelium. Furthermore, in vitro lineage-tracing studies have revealed that the club cell can transdifferentiate to either AEC2 or AEC1 during alveolar generation. Most importantly, club cells form an alveolar-like structure when transforming into AEC2 or AEC1 [31,32]. Strunz et al. [24] showed that in the club cell lineage, cells progress with strong expression of *MHC-II* complex genes, highly similar to alveolar epithelial cells. The resultant *MHC-II^+^* club cell is considered to be the critical gene related to lung epithelial identity development during the *MHC-II^+^* club cell conversion to the intrapulmonary *p63^+^* progenitors.

The above conclusions are based on animal or in vitro experiments. It is possible to study the origins of the transitional state cells in humans using single-cell transcriptomic analyses to make reliable inferences. However, while cell-of-origin between humans and mice may differ, there remain similarities in the dysplastic expansion of transitional state cells across different species. Therefore, results obtained from studies using mice are partially consistent with those of humans.

## 5. The Mechanism of Generation of the Transitional State Cell

Transitional state cell generation relies on multiple pathways, as summarized in Figure 1. During differentiation, the transitional state cell is vulnerable to DNA damage due to excessive stretching, marked by the activation of the *TGF-β* signal pathway, a high expression of cell cycle genes (*Myc* targets), *Wnt/β-catenin*, Nuclear-Factor κB (*NFκB*), and cell senescence. Stress-related signaling pathways are subsequently induced, including the *p53* pathway. Remarkably, genomic binding assays highlight how *p53* directly regulates the transcriptional activity of the transitional state cell. The activation of *TGF-β* is critical for sustaining the transitional cell state [11,24,33,34]. The timely activation of *Myc* plays a critical role in normal cellular proliferation and growth. More importantly, excessive cell proliferation is successfully prevented by the convenient shutdown of *Myc*. Therefore, *Myc* controls whether cells pass through molecular checkpoints successfully. Disturbed molecular checkpoints in lung injury may cause abnormal persistence of the transitional state cell [19,24].

The Hippo signaling pathway plays a critical role during tissue generation after injury. The Yes-associated protein/Transcriptional coactivator with PDZ-binding motif (*YAP/TAZ*) acts as the transcriptional coactivators of *Hippo*, playing a significant role in AEC2-to-AEC1 differentiation during regeneration. From bleomycin-induced mice pulmonary injury models and in vitro organoid culture systems, it can be seen that when *YAP/TAZ* knockout occurs in AEC2, the AEC2 fails to differentiate into AEC1, exacerbating alveolar dysfunction and fibrosis. Although some researchers believe that the *YAP/TAZ* pathway is an alternative for the renewal of normal AEC1 in lung homeostasis, it is critical to maintaining the development and generation of the AEC2-to-AEC1 transition after lung damage [35,36]. Through immunostaining, Strunz et al. [24] found high *YAP/TAZ* activity in *Krt8^+^* ADI cells and a reduced formation of *Krt8^+^* ADI and AEC1 cell states when *Wnt/β-catenin* is inhibited in vitro.

Moreover, hypoxia and inflammation after injury may drive the generation of basal-like cells, one type of transitional state cell. Local lung hypoxia can promote AEC2 differentiation into basal-like cells by inducing hypoxia-induced factor (*HIF1α*), which boosts along the *Notch* signal pathway. Blocking activation of the Notch pathway after deleting *HIF1α* promotes AEC2 migration and differentiation, increasing alveolar recovery quality [19,37]. Choi et al. [22] found that AEC1-like cells strongly upregulate DATP-associated genes rather than mature AEC1 markers in *IL-1β*-treated organoids. They proposed that the *IL-1β* from interstitial macrophages causes AEC2 to converge to DATP via the glycolysis pathway mediated by *HIF1α*, subsequently resulting in the failed conversion of DATPs to AEC1. Therefore, excessive accumulation of these cells and impaired alveolar regeneration arise. It is notable that *HIF1α* deletion in AEC2 prevents the generation of DATPs, finally reducing the rise of AEC1 and alveolar repair. Sustained *IL-1β* treatment causes the gradual accumulation of DATPs and blocks the differentiation of DATPs to AEC1. In addition, Choi et al. found a high expression of markers involved in the glycolysis pathway in DATPs, which inhibits high-glucose metabolism and increases the expression of AEC1 markers while promoting the maturation of AEC1. These considerations suggest inflammation and changes in the cellular environment, partially inducing transitional state cell generation.

It will be necessary in the future to pay increased attention to signaling pathway crosstalk and confirm the intersections. These paths may become underlying targets for new therapeutic measures.

## 6. Fates of Transitional State Cell

It has been well established that AEC2 proliferates and transdifferentiates into AEC1 to reconstruct the alveolar epithelial surface and gas-exchange system. However, the persistence/appearance of transitional state cells after lung fibrosis potentially delays the process of epithelial generation. Therefore, we should investigate the ultimate fate of the transitional alveolar epithelial state cells.

Strunz et al. [30] suggested *Krt8^+^* alveolar epithelial progenitors transit toward AEC1 by using pseudotime analysis and UMAP velocity information, as summarized in Figure 1. They proposed that AEC2 and airway stem cells undergo the same progenitor cell state, eventually differentiating into AEC1. Furthermore, they derived a distinctive key point to control transition timing. As previously mentioned, *TGF-β* was strongly expressed in early differentiation, peaking in intermediate cell states. However, subsequent initiation of late deactivation was seen in vitro, and maturation of the *Krt8^+^* alveolar epithelial progenitors to AEC1 was prompted [27]. Furthermore, differentiation into AEC1 involves several transcriptional regulators, including *Sox4*, *Wnt* signaling pathway beta-catenin (*CTNNB1*), and the transcriptional coactivator TAZ (*Wwtr1*). All of these peak in the transitional alveolar epithelial state cell, hinting that the *TGF-β*, *Wnt*, and *YAP/TAZ* play instructive roles in committing the transitional state cell toward the AEC1 cell [30,35]. For regulating AEC2 destination, bifunctional kinase/RNase--*IRE1α*, the central mediator of the unfolded protein response (UPR) to endoplasmic reticulum (ER) stress, plays a fundamental role in DATP abundance and function. Auyeung et al. [38] demonstrated that the number of DATPs and severity of fibrosis decreases in the bleomycin-induced model from the *IRE1α* knockout or inhibited *IRE1α*. In addition, they clarified that inhibitors of *IRE1α* delay the profibrotic phenotype of DATPs by downregulating the expression of *integrin avβ6*, a key activator of *TGF-β*. Remarkably, there are several non-validated but intriguing hypotheses, such as that proposed by Huang et al. [19], who used pathway enrichment analysis of genes in the *HES1* regulome. They hypothesized that enrichment of the *Notch* signaling pathway in the *HES1* regulome directs the transitional state cell toward a terminal state. Across the entire cycle of cell proliferation and differentiation, the *p53* mediation process was the main mechanism underlying cell cycle arrest. *TP63* may also initiate the transitional cells to a terminal state. This implies that senescence affects the final state of the transitional state cell.

As an intermediate transition cell state, whether cells stay in the transition state permanently or complete their differentiation is particularly significant for the process of fibrosis. Existing research has revealed several possible mechanisms to control the fate of the transitional state cell (shown in Figure 1). These mechanisms may lead to novel therapies to accelerate recovery and suppress fibrogenesis in lung injury.

## 7. Interaction between Transitional State Cells, Macrophages, and Fibroblasts

Cell-to-cell communications are important in maintaining homeostasis and the etiology of diseases. The AEC2 proportion significantly decreases, and the decline in the AEC1 proportion is smaller than that of AEC2 in IPF [39]. Joshi [40] and Cui [41] showed that murine AEC2 cells could recruit monocyte-derived alveolar macrophages. They then provided nutrition and support to fibroblasts in the pulmonary fibrosis model, meaning interactions with macrophages and/or fibroblasts following an AEC2 reduction may accelerate the progression of fibrosis (Figure 2).

Similarly, although monocyte-derived macrophages have been shown to drive fibrogenesis, it is worth investigating whether the transitional state cells interplay with macrophages and fibroblasts. Strunz et al. [30] found that compared with AEC1, *Krt8^+^* alveolar epithelial progenitors display the largest number of receptor–ligand pairs with macrophages and fibroblasts. Additionally, the transitional state cell can form a unique cellular niche, which peaks in the presence of myofibroblasts and macrophages. Alveolar regeneration after bleomycin is induced post-lung injury [24]. However, there is no conclusive evidence to support the effects of the connection between transitional cells and macrophages on the progression of fibrosis. Accordingly, clarifying the effects of the underlying physical mechanisms of macrophages on transitional state cells may contribute to fibrosis treatments.

## 8. Conclusions and Future Perspectives

Pulmonary fibrosis is a progressive disease induced by a malicious cycle of epithelial repair dysfunction after lung injury. Within homeostasis, AEC2 are the major stem cells for alveolar regeneration. Pathogenesis of pulmonary fibrosis is widely thought to arise from the ineffectual regeneration and repaired defects of AEC2. However, its specific mechanism of regenerative dysfunction remains poorly understood. In human and animal studies, an intermediate transition cell related to alveolar regeneration has been confirmed [11]. Epithelial cells have the plasticity to generate cells that may aggravate disease. RNA-seq analysis has revealed that transitional state cells do not exist in healthy lung tissue. These cells serve as the progenitor/stem cells to repair lung damage [12]. The transitional state cells promote AEC1 differentiation and have plasticity toward the AEC2 lineage. However, what is more important is that deficient maturation or dysregulation of the transitional state cells may result in the development or progression of lung disease. AEC1 emanates from AEC2 and other stem cells. They act as the functional cells, covering most of the alveolar surface. The initial presence and proliferation of transitional state cells is not the problem but rather their persistence. After lung injury, defective differentiation of transitional state cells into AEC1 leads to disordered barrier recovery in the lung tissue. It may worsen fibrosis when cells remain in the transitional state [19,23]. During AEC2 differentiation into thin and large AEC1, the transitional cells undergo extensive stretching, which makes them vulnerable to DNA damage, a feature associated with most degenerative pulmonary fibrosis, notably pulmonary fibrosis and cancers. Therefore, many researchers speculated that the transitional cells may be hijacked into cancer programs. Importantly, Choi et al. [22] found that *KRT8^+^ CLDN4^+^* DATP-like cells can be observed in tumors of lung adenocarcinoma patients, which provided the first line of support. Their studies strongly suggested that the correct regulation of the transitional cells is crucial for lung restoration and have the potential to treat lung diseases, even lung cancer. In fact, several articles have revealed that mechanisms underlying cancer development drive tumoral cellular heterogeneity via co-opting regeneration programs [42,43]. However, there is no definitive conclusion that the accumulation of transitional state cells is associated with the cancer, but we believe that the connection between the transitional cells and lung cancer will become a central issue, and we are looking forward to there being more and more scientists paying their attention to this aspect to seek out more potential therapeutics for lung disease, including lung cancer. In addition, it also has been noted that the senescence of transitional state cells may induce fibrotic complications after COVID-19 [20]. Therefore, transitional state cells have the potential to reconstruct the epithelium surface when AEC2 cells are defective.

The severity of pulmonary fibrosis disease is related to the activation and deactivation of the pathways of *TGF-β*, *Notch*, *YAP/TAZ*, and *Wnt/β-catenin*. Activating the signaling pathway may accelerate the conversion of AEC2 to transitional state cells. However, blocking the signaling pathway at an appropriate time to prevent the accumulation of the transitional state cells and tissue fibrosis may be more important. Additionally, by increasing expression of the *HES1* regulome, *TP63* can initiate the transitional cells’ differentiation into AEC1, restoring the well-balanced function of these cells, contributing to the restriction of the fibrogenic loop, and ultimately boosting the alveolus barrier regeneration. In addition, as summarized in Figure 1, dysregulation of the signal pathway has the potential to serve as the therapeutic interference, although this has not been confirmed. These pathways and genes may serve as targets of treatment for pulmonary fibrosis. Therefore, more potential targets should be located to promote the transitional state cells to AEC1 and alleviate lung fibrosis (Figure 2).

## Figures and Tables

**Figure 1 ijms-23-06757-f001:**
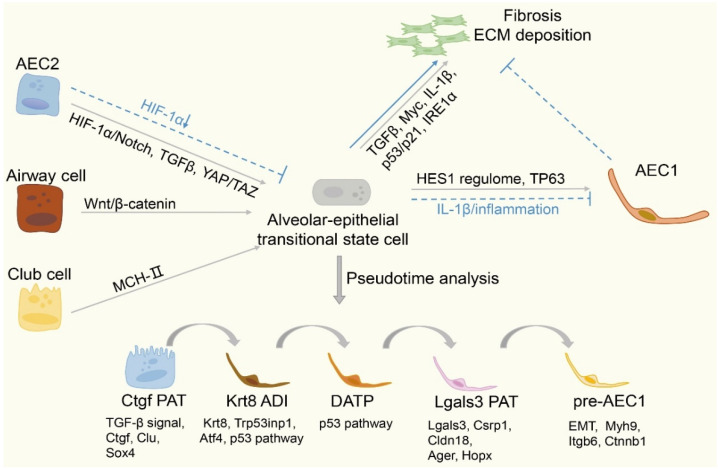
**The differentiation trajectory of the transitional state cell and model of alveolar regeneration.** The transitional state cell can arise from AEC2, airway, and club cells. In this model, the progenitors obtain the highly upregulated *TGF-β*, *Sox4*, *Krt8*, etc., lose their identity markers, and morphologically shift from cuboid to squamous. These cells display cellular senescence-like profiles. Finally, transitional state cells can permanently remain in the transitional state to exacerbate the severity of fibrosis or to complete their proliferation and differentiation to AEC1. The transitional state cells exist in the human and mouse pulmonary model. They serve as checkpoints to control the process of fibrosis.

**Figure 2 ijms-23-06757-f002:**
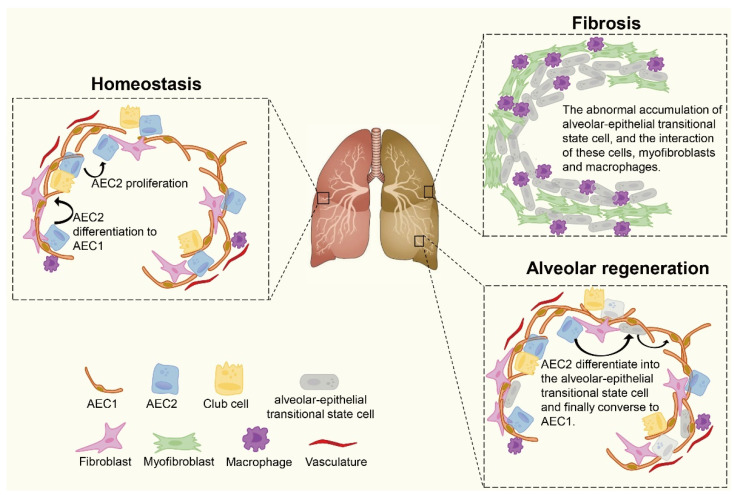
**Cellular composition of lung tissue and alveolar regeneration.** Alveolar cells are mainly composed of AEC2 and AEC1. In homeostasis, most cells have reduced activity, with low AEC2 cell self-renewal and conversion to AEC1 cells. After lung injury, AEC2 increase speed of proliferation and differentiation in order to recover the alveolar barriers. In fibrosis, there is an increase in transitional state cells, accompanied by an increase in myofibroblasts. In addition, these transitional state cells demonstrate the largest communication network with macrophages and fibroblasts. During alveolar generation, AEC2 and the transitional states convert into AEC1 successfully to reconstruct the epithelial barrier and prevent fibrosis.

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
