# Peer review of "Regeneration-Associated Transitional State Cells in Pulmonary Fibrosis"

_ijms, 2022, doi:10.3390/ijms23126757_

Round 1

Reviewer 1 Report

In a fairly extensive review, the authors analyzed the importance of changes taking place within lung cell pools in pulmonary fibrosis. The subject matter is very interesting, as each such summary brings us closer to better treatment of this disease entity.
After reading the work, I believe that the issue of the role of individual types of cells and their metabolism has been treated too marginally. Here it would be appropriate to refer to the applied and potential therapeutic interactions. How the drugs used work, how the side effects are generated, where do they work? The summary that generally talks about possible new therapeutic applications needs to be discussed in detail in the text.
Please, in the last chapter of "7. Conclusion and future perspectives", to propose places of therapeutic interference. Perhaps it would be a good idea to outline a therapeutic intervention pattern.
It is expected that the thesis will be eligible for publication after the extension of the clinical aspects.

Author Response

Dear reviewer,

We deeply appreciate your careful reading and valuable suggestions on our manuscript.  The manuscript has been revised according to your suggestions. 

Point 1: After reading the work, I believe that the issue of the role of individual types of cells and their metabolism has been treated too marginally. Here it would be appropriate to refer to the applied and potential therapeutic interactions. How the drugs used work, how the side effects are generated, where do they work? The summary that generally talks about possible new therapeutic applications needs to be discussed in detail in the text. 

Response 1: Thank you for underlining this deficiency in our manuscript. For the transitional state cell, a novelty cell state, there is no sufficient evidence and research to confirm how to targets it now. One research showed that chronic inflammation mediated by IL-1β prevents AEC1 differentiation, leading to aberrant accumulation of transitional state cell and impaired alveolar regeneration. Their results strongly suggest that the IL-1β-mediated transient inflammatory niche during injury repair is critical for effective lung restoration and is a potential therapeutic adjunct for treating lung diseases1. And for another, researcher clarified that the transitional state cell featured p53 and NF-kB activation and display transcriptional features of cellular senescence, therapeutic approaches may specifically aim at re-programming the transitional state cell into AEC1s by regular p53/p21 pathway2. We summarized in “P4-5, lines 180-226”, as visualized in Figure1. There are many other articles demonstrating the abnormal activation of many pathways in the transitional state cell, however, there is no enough article using signaling pathway inhibitors or agonists to treat pulmonary fibrosis yet. Therefore, this aspect is not described too much in our review. We also hope that our review can attract more researchers pay more attention to the role of transitional state cell in lung fibrosis.

Point 2: Please, in the last chapter of "7. Conclusion and future perspectives", to propose places of therapeutic interference. Perhaps it would be a good idea to outline a therapeutic intervention pattern.

Response 2: Thank you for your valuable comment. We have carefully considered these comments and revised our manuscript in the last chapter of " P8, lines 317-318”. Since there is currently no established conclusion that these signaling pathways can indeed be used as intervention targets, we have not made a detailed explanation in this review. We also hope that this review will be published so that more people can understand the importance of the transitional state cell and find more potential therapy to treat pulmonary fibrosis.

References:

1            Choi, J. et al. Inflammatory Signals Induce AT2 Cell-Derived Damage-Associated Transient Progenitors that Mediate Alveolar Regeneration. Cell Stem Cell 27, 366-382.e367, doi:10.1016/j.stem.2020.06.020 (2020).

2            Strunz, M. et al. Alveolar regeneration through a Krt8+ transitional stem cell state that persists in human lung fibrosis. Nat Commun 11, 3559, doi:10.1038/s41467-020-17358-3 (2020).

Reviewer 2 Report

-Extremely interesting and documented study, with sufficient literature and scientific references of previous studies.

-The etiology and pathogenesis of pulmonary fibrosis are not yet clearly understood so that your extensive and detailed research in this field is crucial in understanding all the mechanisms of the disease and possible treatment strategies.

-The detailed analysis of the transdifferentation of alveolar epithelial type II to epithelial type I and its pivotal role in epithelial repair after lung injury and the possibility that blockage of this process could result in pulmonary fibrosis is expressed in a clear and simple way for the readers. Also the reference and the explanation of the role of transitional state cells and their origin was quite clear too.

-Adequate reference and bibliographic support of the study with previous results from in vivo an vitro studies.

-Very pleasant reading without difficulties in understanding.

Author Response

Dear reviewer:

Thanks for your reading our paper carefully and giving the above positive comments. We are appreciate for your recognition of our work, and we hope that this review can be successfully published to help people better understand the diversity of cells in pulmonary fibrosis disease, and uncover more possible therapeutic targets for lung fibrosis.

Reviewer 3 Report

This study is without novelty because all in it belongs to the previous published article in NAT CELL BIOL 2020 Aug; 22(8):934-946. doi: 10.1038/s41556-020-0542-8. Epub 2020 Jul 13."Persistence of a regeneration-associated, transitional alveolar epithelial cell state in pulmonary fibrosis" by Yoshihiko Kobayashi Y et al.

Author Response

Point 1: This study is without novelty because all in it belongs to the previous published article in NAT CELL BIOL 2020 Aug; 22(8):934-946. doi: 10.1038/s41556-020-0542-8. Epub 2020 Jul 13."Persistence of a regeneration-associated, transitional alveolar epithelial cell state in pulmonary fibrosis" by Yoshihiko Kobayashi Y et al. 

Response 1: Thank you for your important suggestions. The article published in Nat Cell Biol (2020 Aug; 22(8):934-946) is an original article. They made a great contribution to uncovering the transitional state cell in lung fibrosis. In this article, they found a new cell type-transitional alveolar epithelial cell state which is associated with senescence in normal epithelial tissue repair and its abnormal persistence in disease conditions, and they revealed that this transient state cell originates from AEC2. However, our review summarized their work as well as other scientists’ work1-5 as the traits, origins, functions, and activation of signaling pathways of the transitional state cell and its communication with other cells to comprehensively describing and emphasizing the role and characteristics in lung fibrosis.

References

1            Kobayashi, Y. et al. Persistence of a regeneration-associated, transitional alveolar epithelial cell state in pulmonary fibrosis. Nat Cell Biol 22, 934-946, doi:10.1038/s41556-020-0542-8 (2020).

2            Adams, T. S. et al. Single-cell RNA-seq reveals ectopic and aberrant lung-resident cell populations in idiopathic pulmonary fibrosis. Sci Adv 6, eaba1983, doi:10.1126/sciadv.aba1983 (2020).

3            Habermann, A. C. et al. Single-cell RNA sequencing reveals profibrotic roles of distinct epithelial and mesenchymal lineages in pulmonary fibrosis. Sci Adv 6, eaba1972, doi:10.1126/sciadv.aba1972 (2020).

4            Huang, K. Y. & Petretto, E. Cross-species integration of single-cell RNA-seq resolved alveolar-epithelial transitional states in idiopathic pulmonary fibrosis. Am J Physiol Lung Cell Mol Physiol 321, L491-l506, doi:10.1152/ajplung.00594.2020 (2021).

5            Valenzi, E. et al. Disparate Interferon Signaling and Shared Aberrant Basaloid Cells in Single-Cell Profiling of Idiopathic Pulmonary Fibrosis and Systemic Sclerosis-Associated Interstitial Lung Disease. Front Immunol 12, 595811, doi:10.3389/fimmu.2021.595811 (2021).

Round 2

Reviewer 1 Report

I suggest to accept without additional amendments

Author Response

Dear reviewer,

Thanks for your positive comments. We are appreciate for your recognition of our work, and we hope that this review can be successfully published to help people better understand the diversity of cells in pulmonary fibrosis disease, and uncover more possible therapeutic targets for lung fibrosis.

Reviewer 3 Report

The Authors must add the increased risk of cancer due to either the transitional state and the fibrosis that is associated with increased lung cancer risk  as a result of the occurrence of atypical or dysplastic epithelial changes in fibrosis which progressed to invasive malignancy. In that situation, the cancer will develop in the area of major fibrosis.

Every transitional state is at cancer risk.

The Authors must take this into consideration.

The references of this versions are too much and several of minor interest and must be deleted.

Author Response

Dear reviewer,

We deeply appreciate your careful reading and valuable suggestions on our manuscript. The manuscript has been revised according to your suggestions.

Point 1: The Authors must add the increased risk of cancer due to either the transitional state and the fibrosis that is associated with increased lung cancer risk as a result of the occurrence of atypical or dysplastic epithelial changes in fibrosis which progressed to invasive malignancy. In that situation, the cancer will develop in the area of major fibrosis. Every transitional state is at cancer risk. The Authors must take this into consideration.

Response 1: Thank you for underlining this deficiency in our manuscript. In our previous manuscript, we have discussed that the transitional state cells undergo extensive stretching during AEC2 differentiation into thin and large AEC1, which makes them vulnerable to DNA damage, however, we didn’t describe they may increase lung cancer risk. Kobayashi.et al revealed that this state cell is associated with most degenerative lung diseases, notably pulmonary fibrosis and cancers1,2. So we added this aspect in page 7, lines 303-309.

Point 2: The references of this versions are too much and several of minor interest and must be deleted.

Response 2: Thanks for your valuable comment. We have deleted the unnecessary references in our new version manuscript. However, we still stayed several important references which are related to lung fibrosis, epithelial cells and signal pathways.

Reference:

1            Kobayashi, Y. et al. Persistence of a regeneration-associated, transitional alveolar epithelial cell state in pulmonary fibrosis. Nat Cell Biol 22, 934-946, doi:10.1038/s41556-020-0542-8 (2020).

2            Ruaro, B. et al. The History and Mystery of Alveolar Epithelial Type II Cells: Focus on Their Physiologic and Pathologic Role in Lung. Int J Mol Sci 22, doi:10.3390/ijms22052566 (2021).

Round 3

Reviewer 3 Report

This reviewer thinks that the Authors only partially answered to the previous concerns. They only added some words in a paragraph and regarded superficially this important aspect of cancer risk.

Moreover, the specific citations that they deleted and their numbers are not specified.

Their fast corrections are not sufficient.

They must be more precise and study in deep the topic

Author Response

Point 1: This reviewer thinks that the Authors only partially answered to the previous concerns. They only added some words in a paragraph and regarded superficially this important aspect of cancer risk.

Response 1: Thanks for your important suggestion. Choi et al.1 reported that they observed the transitional state cell in tumor of lung adenocarcinoma patients, which provided the first line of support that transitional state cell may have cancer risk. However, although they observed the presence of this cell in tumor tissue, the connection between the transitional state cell and lung cancer was not further elucidated. Furthermore, mechanisms underlying cancer development drive tumoral cellular heterogeneity via co-opting regeneration programs2,3. Based these informations, many scientists speculated that cells in transitional state may be hijacked into cancer programs4-6. In our new version manuscript, we discussed it and expressed our desire in P7-8, lines 306-318, marked in red. Unfortunately, there is no more other references to support this opinion, this is also a shortage of this review. In the future, we hope our review will arouse other scientist interests, and investigate the connection between the transitional state cell and progress of cancer, for the purpose of seeking out more potential therapeutics for lung disease, including lung cancer. 

Point 2: Moreover, the specific citations that they deleted and their numbers are not specified. 

Response 1: We are so sorry that we didn’t indicate the deleted citation in our previous manuscript. We deleted No.2-3 references in the introduction, but failed to mark it due to our negligence. And in our new version manuscript, we also add few references in discussion and we marked it.

 References:

1            Choi, J. et al. Inflammatory Signals Induce AT2 Cell-Derived Damage-Associated Transient Progenitors that Mediate Alveolar Regeneration. Cell Stem Cell 27, 366-382.e367, doi:10.1016/j.stem.2020.06.020 (2020).

2            Moll, H. P. et al. Afatinib restrains K-RAS-driven lung tumorigenesis. Sci Transl Med 10, doi:10.1126/scitranslmed.aao2301 (2018).

3            Maynard, A. et al. Heterogeneity and targeted therapy-induced adaptations in lung cancer revealed by longitudinal single-cell RNA sequencing. bioRxiv, 2019.2012.2008.868828, doi:10.1101/2019.12.08.868828 (2019).

4            Verheyden, J. M. & Sun, X. A transitional stem cell state in the lung. Nat Cell Biol 22, 1025-1026, doi:10.1038/s41556-020-0561-5 (2020).

5            Kobayashi, Y. et al. Persistence of a regeneration-associated, transitional alveolar epithelial cell state in pulmonary fibrosis. Nat Cell Biol 22, 934-946, doi:10.1038/s41556-020-0542-8 (2020).

6            Huang, K. Y. & Petretto, E. Cross-species integration of single-cell RNA-seq resolved alveolar-epithelial transitional states in idiopathic pulmonary fibrosis. Am J Physiol Lung Cell Mol Physiol 321, L491-l506, doi:10.1152/ajplung.00594.2020 (2021).

Round 4

Reviewer 3 Report

Now the Authors have answered to my previous criticisms.